# The Significant Impact of Carbon Nanotubes on the Electrochemical Reactivity of Mg-Bearing Metallic Glasses with High Compressive Strength

**DOI:** 10.3390/ma12182989

**Published:** 2019-09-15

**Authors:** Jing-Feng Lou, Ai-Guo Cheng, Ping Zhao, R. D. K. Misra, Heng Feng

**Affiliations:** 1State Key Laboratory of Advanced Design and Manufacture for Vehicle Body, Hunan University, Changsha 410082, China; loujingfengqdu@163.com; 2College of Materials Science and Engineering, Qingdao University of Science & Technology, Qingdao 266044, China; zhaoping1986711@163.com; 3Department of Metallurgical, Materials and Biomedical Engineering, University of Texas, El Paso 500 W. University Avenue, El Paso, TX 79968-0520, USA; dmisra2@utep.edu; 4SAIC-GM-Wuling Automobile Co., Ltd., Liuzhou 545007, China; fengheng1232002@hotmail.com

**Keywords:** bulk metallic glasses, CNTs, electrochemical activity, mechanical properties

## Abstract

Here, we elucidate the significant impact of carbon nanotubes (CNTs) on the electrochemical behavior of Mg-based amorphous composite materials that were reinforced with CNTs while using pressure die casting. The addition of 3 vol % CNTs led to an increase in the compressive strength of Mg-based amorphous material from 812 MPa to 1007 MPa, and the fracture strain from 1.91% to 2.67% in the composite. Interestingly, the addition of CNTs significantly contributed to the enhancement of corrosion resistance of Mg-based glass by ~30%. The superior mechanical properties are primarily related to the fact that the addition of CNTs hindered the growth of shear bands (cracks), while the high corrosion resistance is related to inferior wettability and the bridging effect between adherent corrosive oxide film and the matrix that provided enhanced corrosion resistance.

## 1. Introduction

Amorphous alloys are being given increased attention since their birth due to their high strength, high hardness, and high corrosion resistance, etc. Among different amorphous alloys, Mg-based bulk metallic glasses (BMGs) are relatively new materials for various engineering applications due to their superior mechanical properties and corrosion resistance [1,2]. Recently, BMGs attracted significant attention as a new class of biodegradable materials [3]. Some of their properties (e.g., strength, hardness, etc.) have been observed to be superior as compared to their crystalline counterparts. Numerous Mg-based amorphous alloys or BMGs have been developed since the preparation of Mg_70_Zn_30_ metallic glass in 1977 [4]. Until now, majority of the glass-forming compositions are based on Mg-TM-RE or Mg-TM-Ca ternary systems (where TM = transition metals Cu, Ni, Zn, Ag, and RE = rare earth metals Y, Gd, La, Nd, Ce, etc.) [5,6]. Some of these alloys have exceptional glass-forming ability with a critical casting size of up to 27 mm [7].

However, the Mg-based amorphous alloy is still limited in engineering applications. Essentially, the key issue precluding the use of Mg-based BNGs for at least small-scale structural applications is their inherent low fracture toughness or brittle nature. This shortfall in mechanical performance is due to (i) the low activation energy for shear band formation and (ii) rapid structural relaxation effects of Mg-based glasses [8]. Thus, severe embrittlement can occur, even at room temperature and at relatively short time scale strongly because of the low glass transition temperatures (*T*_g_).

Another factor limiting the commercial application of Mg-based BMGs is their relatively low corrosion resistance [9,10]. In recent years, many studies have been carried out on the corrosion behavior of Mg-based BMGs. Gebert et al. [11,12] showed that Mg-Cu-Y amorphous alloy had higher corrosion resistance than the traditional Mg alloy, where the addition of Ag was beneficial in improving the corrosion resistance of Mg amorphous alloy [12]. The corrosion behavior of Mg-Cu-Y and Mg-Ni-Nd amorphous alloys was analyzed by Yao et al. [13], who observed that the presence of NiO, Nd_2_O_3_ (Mg-Ni-Nd) and CuO, Y_2_O_3_ (Mg-Cu-Y) in the passive films of the two alloys, which contributed to corrosion resistance. Qin et al. [14] studied the corrosion properties of Mg_65_Cu_25_Gd_10_ amorphous alloys and their corresponding crystalline alloys. In summary, the new Mg-based alloy glasses require not only good mechanical properties, but also good corrosion resistance to stand the environment.

In order to improve the mechanical properties and corrosion resistance of Mg-based BMGs, it is an available approach to add a second phase with excellent strength and elastic modulus into an amorphous matrix to avoid the formation of highly localized shear bands, and promote the slip of multiple shear bands formation, which can effectively improve the toughness of the amorphous alloy. In addition, it can increase the corrosion potential, reduce the corrosion current density, and enhance corrosion resistance. Carbon nanotube (CNT) is undoubtedly an ideal choice amongst the second phase materials that can be added to Mg-based glasses. It has attracted worldwide interest in the past two decades, owing to its wide applications in numerous fields [15]. CNT with sp^2^ hybrid C-C bond is the strongest chemical bond, which renders it with extraordinary mechanical strength. For example, its Young’s modulus is higher than 1 TPa [15,16], and its tensile strength is higher than 100 GPa [17]. The weight-specific strength of the CNT is at least four hundred times higher than the steel. 

In recent years, studies have been carried out on metal matrix composite containing CNTs [18,19]. Wang et al. [20,21] studied the effect of CNTs on the performance of Zr-based amorphous alloy. However, studies on the impact of CNTs in bulk amorphous alloy are still limited, especially the effect on fracture mechanisms and corrosion resistance in Mg-based BMGs. In this study, we have prepared the amorphous alloy composites by adding CNTs in Mg-based BMGs and elucidated their influence on the mechanical properties of Mg_60_Cu_2__3_Gd_11_Ag_6_ bulk metallic glass. Besides, we study the improvement mechanism of corrosion properties of Mg-BMG by CNTs, which can provide theoretical support for the application of Mg-BMG in medicine.

## 2. Experimental Procedure

### 2.1. Materials

The purity of Mg and Cu used as raw materials was greater than 99.99%, and the purity of Gd and Ag was ~99.9%. Multiwalled CNTs with more than 95% purity were used as the reinforcement phase, which were prepared by a chemical vapor deposition method. The CNTs were pre-treated before adding to the amorphous alloy in order to uniformly disperse them and avoid agglomeration in amorphous alloys. The pre-treatment involved ultrasonication in acetone and dehydration at 200 °C. The morphology and structure of CNTs was observed by transmission electron microscope (TEM, FEI TECNAI G20, Eindhoven, Netherlands, 200 KV) and it is presented in Figure 1. The diameter of CNTs was in the range of 20–30 nm and the length of several micrometers.

### 2.2. Preparation Process of Metallic Glass

The method of preparation of Mg_60_Cu_23_Gd_11_Ag_6_ bulk metallic glass was as follows: (1)Cu-Gd-Ag intermediate alloy was prepared by arc-melting under a Ti-gettered argon atmosphere in a water-cooled copper container, as shown in Figure 2a. The button-like alloy ingot was obtained after melting the intermediate alloy while using vacuum arc melting. Each ingot was re-melted four times to ensure a homogeneous composition.(2)Next, Mg was mixed with the intermediate, Cu-Gd-Ag alloy, based on the atomic ratio of the final alloy. The crystalline Mg-Cu-Gd-Ag master alloy was obtained by induction melting in an electromagnetic induction furnace. Figure 2b shows the internal view of electromagnetic induction melting furnace.(3)In the third stage, the master alloy was ball milled (Figure 2c) to obtain small particles, and then mixed with the desired content of CNTs. The ball milling was carried out for 2–6 hours, and pressed as a pellet.(4)The mixed pellet was placed in a high-temperature-resistance stainless steel crucible, which was melted by electromagnetic induction, and the liquid metal was rapidly pressed in a water-cooled copper mold while using a hydraulic ejector rod. The liquid metal was rapidly condensed into metallic glass. Figure 2d shows the water cooling die casting system with copper mold.

A large number of experiments confirmed that the amorphous metallic materials with excellent properties can be successfully prepared by the above-mentioned process [22,23,24]. In this study, two types of amorphous alloy materials were prepared, namely CNTs-free Mg_60_Cu_23_Gd_11_Ag_6_ metal glass, abbreviated as Mg-BMG, and 3 vol % CNTs containing Mg_60_Cu_23_Gd_11_Ag_6_ metal glass, abbreviated as CNTs-Mg-BMG.

### 2.3. Methods for Testing the Properties of Experimental Materials

The samples were analyzed by X-ray diffractometer (D/max-RB, Tokyo, Japan) to determine whether the alloy was amorphous. The copper target Kα radiation, λ = 1.5418 Å, voltage 40 kV, current 200 A, and scanning speed 8 °/min. were used for analysis. Uniaxial compression test is one of the most direct and effective methods to test mechanical properties. In the experiment, the compression experiment at room temperature was carried out on a AGS-10KNG universal material testing machine, and the minimum compression speed of the test equipment was 0.05 mm/min. The sample for compression test was a cylindrical sample of diameter 2 mm and aspect ratio 2:1, which was cut by a diamond saw. The axis direction of the sample rod was perpendicular to the cutting direction, and the two ends of the sample were polished after cutting to reduce the influence of the surface [25]. The strain rate of compression test was 3 × 10^−4^s^−1^ [26]. Fracture surface was examined while using FEI Quanta 200 (FEG) scanning electron microscope (SEM) (FEI, Eindhoven, Netherlands) [27]. 

The two kinds of experimental BMG materials of Φ4×3 mm dimensions were completely immersed in 0.1 mol/L NaOH solution for 120 h [28,29]. Next, the sample was immersed in 200 g/L CrO_3_ solution for 5 min. to remove the corrosion product on the surface of the sample, and then washed with distilled water and weighed after drying.

## 3. Results

### 3.1. Amorphous Structure

Figure 3 shows images of water-cooled copper die cast glassy Mg_60_Cu_23_Gd_11_Ag_6_ BMG and composites with 3 vol % CNTs, together with the XRD patterns that were obtained from the cross-sectional surface of the rod of BMG and composite materials. From Figure 3a, it can be seen that the die casting amorphous alloys of 2–5 mm diameter had a bright and shining appearance, which is characteristic of amorphous alloys. In Figure 3b, the absence of sharp crystalline diffraction peaks and the presence of a broad diffraction peaks at 2θ = 35° implied the formation of glassy phase of Mg-BMG. For CNTs-Mg-BMG composite, the XRD pattern indicated a superimposition of a broad maximum of the amorphous structure and one or two sharp peaks characteristic of crystalline phase, which suggests a mixed structure of amorphous and some crystalline phases. The 2θ position of the crystalline peaks corresponded to the Gd_2_C_3_ phase (Figure 3b). No other phases were detected within the sensitivity limit of XRD. The results indicated that some of the CNTs reacted with Mg/Gd and formed crystalline Gd_2_C_3_ phase. The Mg and Gd elements have a large negative heat of mixing with C, in comparison to other constituents (Cu, Ag), which ipmplied that Mg/Gd and C have a large driving force for the interfacial reaction.

### 3.2. Effect of CNTs on Mechanical Properties of Mg Alloy Glass

Figure 4 is the quasi-static compressive stress-strain plots of two BMGs at room temperature. It can be seen from figure that there are only elastic regions of deformation. For the Mg_60_Cu_23_Gd_11_Ag_6_ BMG sample, a classical amorphous elastic appearance was observed, with predominantly elastic behavior. However, the compressive stress-strain plots of CNTs-BMG composite indicated some differences with respect to the non-CNT reinforced BMG, i.e., the compressive strength was prodigiously increased, exceeding 1000 MPa, which was more than three times of the crystalline magnesium-based alloy. This result implies that the addition of CNTs led to dispersion strengthening and second phase strengthening [30]. 

From Figure 4, it is noted that the compressive stress-strain curves of the two experimental amorphous materials presented a saw-tooth type of profile, and each saw tooth flow represented the formation of a shear band [31]. The shear band began to move on formation, which released part of the elastic deformation energy, which resulted in the decrease of macroscopic stress, and the formation of saw tooth morphology in the stress-strain curve.

Table 1 summarizes the mechanical and physical properties. From the table, it may be noted that the compressive strength increased from 812 MPa of Mg-BMG to 1007 MPa of CNTs-Mg-BMG, respectively. The fracture strain increased from 1.91% of Mg-BMG to 2.67% of CNTs-Mg-BMG, respectively. With the addition of CNTs, the density of material was decreased, and the compressive strength per unit mass was increased from 188 kN·m·kg^−1^ of Mg-BMG to 245 kN·m·kg^−1^ of CNTs-Mg-BMG. It can be concluded that the improvement of mechanical and physical properties is due to the high specific strength and high ductility of CNTs.

Figure 5 is the compression fracture morphology of Mg-BMG that was observed by SEM. Under the compressive stress, Mg-BMG produced localized shear bands, as shown in Figure 5a. Unlike other bulk amorphous alloys, there is no specific angle between the fracture surface and loading direction after fracture in compression test, and the shear bands are nearly parallel to one another. The adiabatic heating occurred during compressive deformation of amorphous alloy, and the increase of local temperature of amorphous alloy during deformation can be estimated by the following Equation (1) [32,33]:
(1)$$\Delta T=\frac{k_{c}\sigma_{y}\left({1-\upsilon }^{2}\right)\sqrt{2V}}{E\sqrt{\rho CK}}$$
where *ρ* is the density, *C* is the thermal capacity, *K* is the thermal conductivity, and *V* is the crack propagating speed. The Δ*T* was calculated as 420–470 K from the physical parameters of the Mg-based and CNTs-Mg-based BMG, as measured in the literature [32]. The *T*_g_ and *T*_m_ of the Mg-based BMG were determined as 416 K and 682 K, respectively, and the two temperatures of another BMG as 422 K and 678 K. The calculated Δ*T* was between *T*_g_ and *T*_m_. The local temperature will rise when the BMG is subjected to a compression stress from the literature [22]. Local plastic deformation will not occur if the temperature is lower than the melting temperature.

From Figure 5b, it can be seen that the compressive fracture surface of Mg-BMG was relatively smooth, and shear bands were in the same direction, where the distance between the two shear bands was ~50 μm. Figure 5c is the magnified view of Figure 5a. It can be seen from Figure 5c that there were secondary cracks between shear bands, which indicated that shear bands were prone to cracking [15]. Mg-based metal glass showed typical brittle fracture morphology of amorphous alloys.

The fracture characteristics of Mg-BMG significantly changed after adding CNTs. Figure 6 shows that the fracture morphology of CNTs-Mg-BMG. It may be noted from the figure that shear bands disappeared on encountering CNT cluster, which indicated that the presence of CNT cluster hindered the propagation of shear bands and cracks. From Figure 6b, the CNTs were easily aggregated together to form a CNT cluster, and they were wrapped around the alloy so that their size was much larger than the diameter of individual CNT. Dimples were observed on the fracture surface of size ~300 nm. The relatively flat vein pattern at the perimeter of the shear band (or crack) was clearly observed. From Figure 6c, we can see that there were large vein-like patterns in the fracture surface, where the distance between vein patterns was several nanometers. The appearance of vein pattern is related to the increase of temperature, localized softening, and internal melting of shear bands before compressive fracture [22,34]. 

The mechanical properties of the material were closely related to the microstructure. When CNTs were not added, the Mg-BMG structure of 2 mm bar was completely amorphous. The deformation under uniaxial stress at room temperature was concentrated in one or more shear bands. The growth and extension of shear bands was not hindered by any obstacles, so they can pass through the entire specimen section quickly, which results in brittle rupture of the specimen in the elastic deformation stage. In the case of CNTs-Mg-BMGs, the Mg alloy powder and CNTs were ball milled (Section 2.2) and the CNTs were well bonded to the alloy, and some crystal phases, including Gd_2_C_3_, formed in the amorphous alloy. These phases, together with CNT cluster, would play an important role in hindering crack growth [35]. In addition, the mechanical properties of CNTs are superior to the Mg alloy, which can restrict the extension of shear bands (Figure 6a,b), such that the mechanical properties of the CNTs-Mg-BMG were superior to the Mg-BMG.

### 3.3. Corrosion Resistance

The corrosion rate of the two experimental materials was calculated while using the following Equation (2) [36]:(2)$$R_{\mathrm{corr}}=\frac{W_{0}{-W}_{1}}{At}\times 100\%$$
where *W*_0_ and *W*_1_ represent the quality of the samples before and after corrosion, respectively. *A* represents the surface area of BMGs, and *t* is the time of corrosion. Three samples were tested and the average value of corrosion rate was obtained. The corrosion rate of Mg-BMG in NaOH solution was 0.96 mg/cm^2^·d, while the corrosion rate of CNTs-Mg-BMG in NaOH solution was 0.66 mg/cm^2^·d. Thus, the corrosion resistance of CNTs-Mg-BMG was superior by ~30% than Mg-BMG, which implied that CNTs improved the corrosion resistance of BMG. 

Figure 7 shows the corrosion morphology of Mg-BMG after soaking in 0.1 mol/L NaOH solution for 120 h at room temperature. The amorphous alloy experienced severe corrosion and the surface of the corroded amorphous alloy was rough, which was a typical corrosion morphology of Mg alloy amorphous [9]. Based on Figure 7a, the corrosion layer peeled off in Mg-BMG corroded surface, and there were a number of cracks beneath the peeled layer. Figure 7b is a magnified view of the severely corroded area, we can see corrosion pits of size ~20 μm, and many small corrosion pits of size 1–2 μm were present inside the large corrosion pit after corrosion. Moreover, from Figure 7c, there were a large number of particles in the corrosion pits, and nanoscale size pores between the particles.

Figure 8 shows the corrosion morphology of amorphous composites with 3vol % CNTs in NaOH solution. From Figure 8a, it can be seen that the surface of CNTs-Mg-BMG sample was relatively flat after corrosion, and corrosion pits were small, of size 1–5 μm. The thickness of the corrosion peeling layer was small, and in some places there was only a small difference between the uncorroded and corroded surface. Figure 8b is the magnified view of the corroded region. There were snowflake-like crystals in the corrosion pits, and a number of small corrosion cracks, similar to the intergranular corrosion fracture mechanism between the snowflake crystals. The corrosion morphology was further magnified, as shown in Figure 8c. It can be seen that the grain size of snowflake-like area corrosion was uneven, from ~500 nm to ~2 μm. Meanwhile, many grains were agglomerated to form aggregates, and CNTs were present in aggregates. Additionally, the alloy wrapped CNTs. There was an intergranular gap (corrosion crack) between the snowflake-like corrosion grains, and the gap size was ~500 nm.

The corrosion products were analyzed by energy dispersive spectrometry (EDS), as presented in Figure 9. It can be seen that the peak value of oxygen was relatively high, suggesting that the corrosion product consisted of oxides. Table 2 lists the atomic content of each element. According to Figure 9 and Table 2, we can conclude that the corrosion products were Mg(OH)_2_ and Cu(OH)_2_.

## 4. Discussion

### 4.1. Shear Deformation Resistance Mechanism of CNTs

CNT is a nanomaterial with large aspect ratio, with a strengthening mechanism that is similar to fiber reinforcement. It may be seen from Figure 6 that the interface between CNTs and amorphous matrix was good and the bond strength was also good. When the composite amorphous material was formed by rapid cooling, CNTs with large elastic modulus were tightly wrapped by the alloy. During compressive stress, a similar “reinforced concrete” effect can be imagined, which hindered the expansion of shear bands (Figure 6). 

In our experiments, we used multi-walled CNTs, where the inner and outer carbon tubes can easily slip, because of small inter-layer interaction force between them. Under stress, the outer layer of CNTs can collapse to form a shear stress zone and absorb the fracture energy of matrix. Meanwhile, the inner CNT layer is pulled out and peeled off. The mutual slip of the inner and outer carbon tube layer can absorb the stress of the matrix, contributing to an enhancement effect. The enhanced effect of CNTs on Mg-based BMG is represented in Figure 10. According to the shear-lag theory [37], the interaction between the CNT and the matrix that resulted from the chemical bonds was shear stress, and was related to the relative displacement between the CNT and the matrix, i.e., Δ*μ*. The bond is broken when Δ*μ* reaches critical shear displacement, i.e., *δ_b_*, which only depends on the type of functionalization bond at the interface, whereas the corresponding interface strength *τ_b_* also depends on the interface bond density [38]. The interface shear stress τ is assumed to be proportional to the relative displacement Δ*μ,* according to Equation (3) [39]:
*τ*(*x*) = *k*Δ*μ*(*x*) = *k*[*μ*_m_(*x*) − *μ*_f_(*x*)](3)
where *k* = *τ_b_*/*δ_b_* is the shear stiffness of the interface and *μ_m_*(*x*) and *μ*_f_(*x*) are the axial or displacements in the *x* -axis direction of matrix and CNT, respectively. 

The CNT and the matrix both experienced linear elastic behavior, with Young’s modulus *E*_f_ and *E*_m_, respectively. A representative volume element (RVE), including a single CNT with embedded length L and diameter d, as shown in Figure 10, was adopted for analysis. With the balance conditions of the CNT and the matrix, the shear stress distribution can be derived, as follows Equation (4) [40]:
(4)$$\tau (x)=F\cdot \frac{t}{C}\cdot \frac{\mu_{m}\cos h\left(xt\right){+\mu }_{f}\cos h\left[t\left(x-t\right)\right]}{\sin h(Lt)}$$
where *μ*_m_ =$\;\frac{1}{E_{m}A_{m}}$, *μ*_f_ = $\frac{1}{E_{f}A_{f}}$, *t* = $\sqrt{\frac{{C\tau }_{b}}{\delta_{b}}}$, *A*_f_ and *A*_m_ are the cross-section areas of the CNT and the matrix in the RVE, *F* is the pulling force, and *C* depends on the material constants and geometry parameters, as follows Equation (5):*C* = π*d*(*μ*_m_ + *μ*_f_)(5)

The distribution of the axial normal stress in the CNT can also be derived, as follows, Equation (6):(6)$$\sigma x\;\frac{1}{A_{f}}\left[F-\int_{0}^{x}\pi d\tau \left(x\right)dx\right]$$

Given that the elastic modulus and tensile strength of CNTs are significantly higher than the matrix alloy, the failure mode of CNTs is mainly interfacial debonding. For CNT-reinforced composites, the maximum shear stress at the interface can be obtained at the *x* = *L* position, and the maximum shear stress is Equation (7) [40]:(7)$$\tau (x){|}_{\max}=\tau \;(L)=F\cdot \frac{t}{C}\cdot \frac{\mu_{m}\cos h\left(Lt\right)\;{+\mu }_{f}}{\sin h(Lt)}$$

When the maximum shear stress reached the interface strength *τ_b_*, the interfacial chemical bonds break, and the CNT slipped and collapsed. The corresponding critical pulling force is illustrated in Equation (8):
(8)$$F_{max}^{\tau }=\frac{C}{t}\cdot \frac{\tau_{b}sinh(Lt)}{\mu_{m}cosh\left(Lt\right){+\mu }_{f}}$$

It can be seen from the above equations that the contribution of CNTs to Mg-BMG mainly depends on the binding force between CNT and the metal matrix, which increases with the increase of bonding force.

### 4.2. The Effect of CNTs in Improving the Corrosion Performance of Mg-BMGs

In corrosive medium, Mg alloy matrix and solution ions can easily form corrosion battery and induce galvanic corrosion because of the high chemical activity of the Mg alloy. In addition, according to metallic corrosion principle [41], the oxide film has a protective effect when the volume of the oxide film formed during the oxidation process of the metallic material is larger than the metal consumed by the formation of the oxide films. However, the density coefficient of the oxide film formed on the surface of Mg alloy is 0.84 [42]. Thus, the oxide film is not dense and it cannot form a stable protective film. Meanwhile, with the evolution of a large amount of hydrogen during the corrosion of the Mg alloy, small hydrogen bubbles are generated. The formation, enrichment, and diffusion of hydrogen bubbles can result in the loosening of corrosion product film, reducing the adhesion between the corrosion film and the Mg alloy matrix, which leads to the peeling of local corrosion product film from the substrate (Figure 7), which is the main reason for the inferior corrosion resistance of Mg alloy. On adding CNTs to Mg-BMG, there is portion of CNTs on the surface of the composite, where the wettability of matrix and water is reduced to some extent, because CNTs have very strong corrosion resistance and poor wettability with water. Thus, the corrosion resistance of CNTs-Mg-BMG composite is improved. In addition, CNTs dispersed in the metal matrix bridged the cracks in the oxide layer between the oxide layer and the matrix, thereby hindering the formation of corrosion cracks in the oxide layer on the surface of the composite, and preventing the oxide layer peeling of the matrix. As shown in Figure 8, the peeling layer of CNTs-Mg-BMG was smaller than Mg-BMG. In this way, the penetration of corrosion medium into the matrix was hindered and further corrosion of composite was delayed. Thus, the addition of CNTs enhanced the corrosion resistance of Mg-based glasses.

## 5. Conclusions

In this study, the CNTs-Mg based amorphous composites were successfully prepared by water cooled copper die casting. The effect of CNTs addition on the properties of Mg based metallic glass was studied by uniaxial compression test and full immersion corrosion test. The main conclusions are as follows: (1)On adding 3 vol % CNTs, the compressive strength was raised by 812 MPa of Mg-BMG to 1007 MPa of CNTs-Mg-BMG, and increase of ~24%.(2)The corrosion resistance of amorphous matrix in 0.1 mol/L NaOH solution was improved by the addition of CNTs, where the corrosion rate decreased from 0.96 mg/cm^2^·d of Mg-BMG to 0.66 mg/cm^2^·d of CNTs-Mg-BMG, and a decrease of ~30%.(3)Under the action of compressive stress, the damage of the binding force between the CNTs and the metal matrix can absorb the stress deformation energy, such that the compressive stress is shared and the compressive strength enhanced.(4)CNT improved the corrosion performance because of its high corrosion resistance and poor wettability, and it included the bridging effect between corrosive oxide film and the matrix, which made the oxide film difficult to peel-off, thereby improving the corrosion resistance.

## Figures and Tables

**Figure 1 materials-12-02989-f001:**
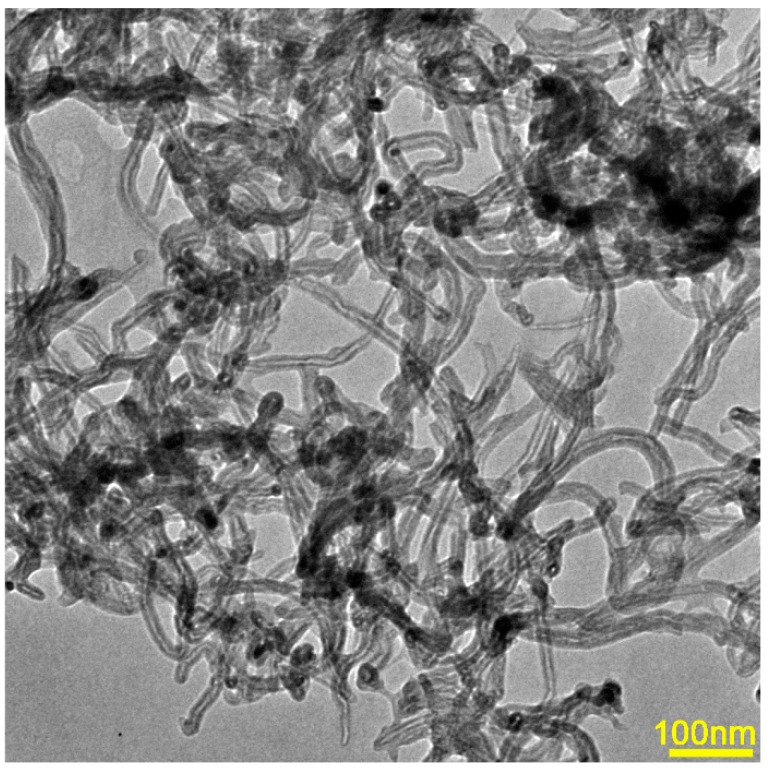
Morphology of carbon nanotubes observed by transmission electron microscope.

**Figure 2 materials-12-02989-f002:**
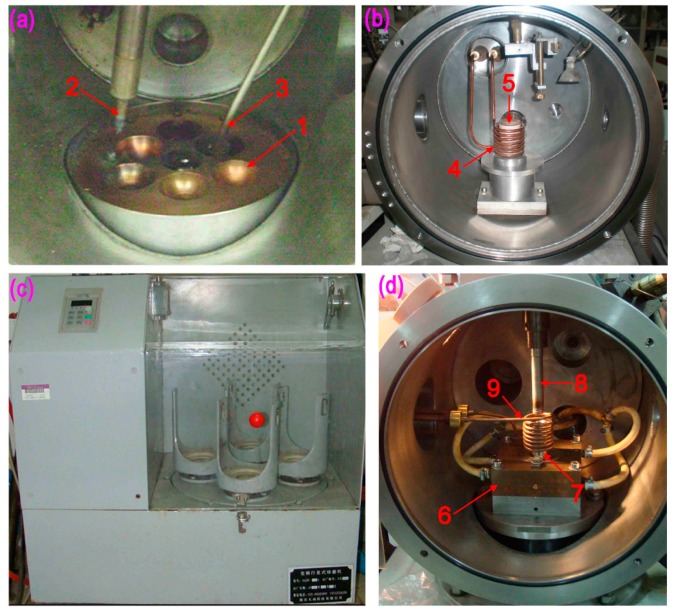
Preparation equipment of carbon nanotubes reinforced Mg alloy glass: (**a**) Electric arc furnace for melting Cu-Gd-Ag intermediate alloy; (**b**) electromagnetic induction furnace for melting Mg-Cu-Gd-Ag master alloy; (**c**) ball milling of carbon nanotubes and master alloy particles for 2–6 h; (**d**) water cooling die casting system with copper mold, where the mixture material is heated to a high temperature in stainless steel crucible, and rapidly pressed into a water-cooled copper die to obtain glassy state. (1—Copper based crucible; 2—Arc electrode; 3—Manipulator for turning ingot; 4—Induction coil made by copper tube with circulating water; 5—Ceramic crucible; 6—Copper mould; 7—Crucible made by high temperature stainless steel; 8—Pressure casting rod; and, 9—Induction coil made by copper tube with circulating water).

**Figure 3 materials-12-02989-f003:**
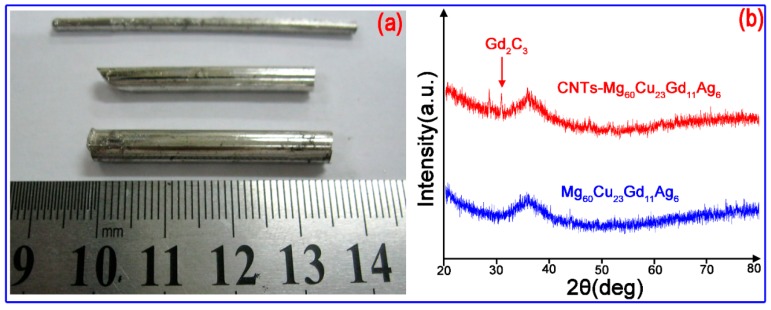
(**a**) Appearance of amorphous alloy rods with different diameters; and (**b**) X-ray diffraction patterns of two bulk metallic glasses.

**Figure 4 materials-12-02989-f004:**
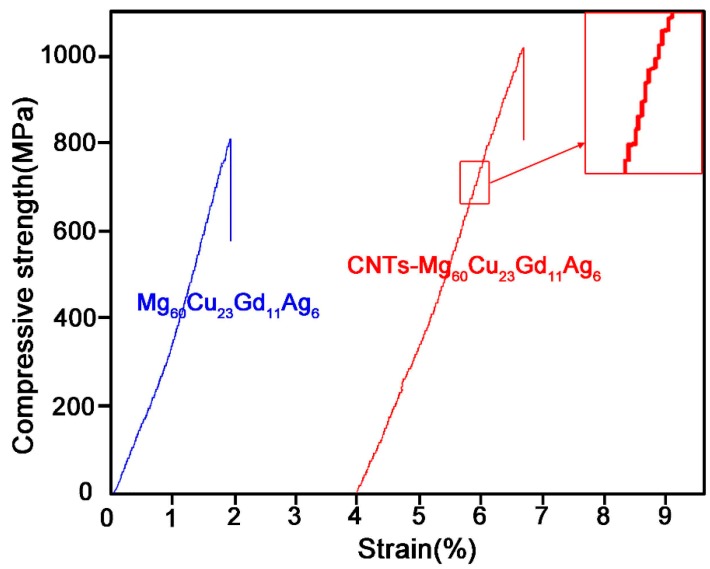
Comprehensive stress-strain curves of Mg-based bulk metallic glass and carbon nanotubes reinforced Mg-based composite glassy rods.

**Figure 5 materials-12-02989-f005:**
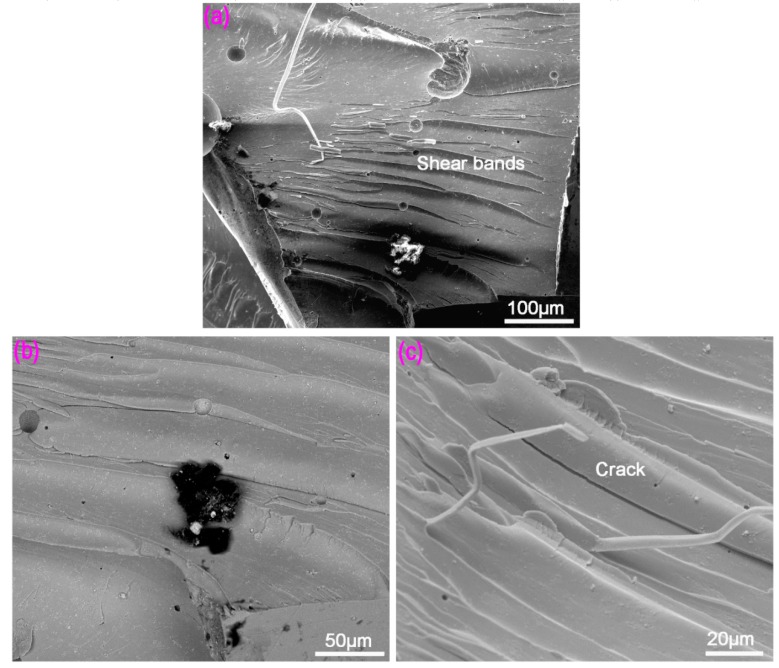
Fracture surface of copper mold die cast amorphous alloy of Mg_60_Cu_23_Gd_11_Ag_6_: (**a**) Overall morphology of compression fracture surface; (**b**) local melting zone morphology; and (**c**) shear bands and cracks.

**Figure 6 materials-12-02989-f006:**
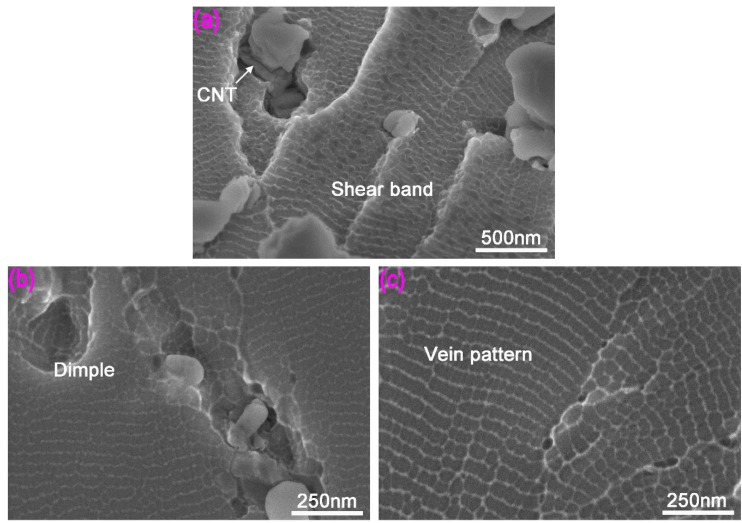
Fracture surface of copper mold die cast amorphous composite material of carbon nanotubes reinforced Mg_60_Cu_23_Gd_11_Ag_6_: (**a**) Morphology of compressive fracture shear band; (**b**) dimples and crack; and (**c**) vein pattern morphology.

**Figure 7 materials-12-02989-f007:**
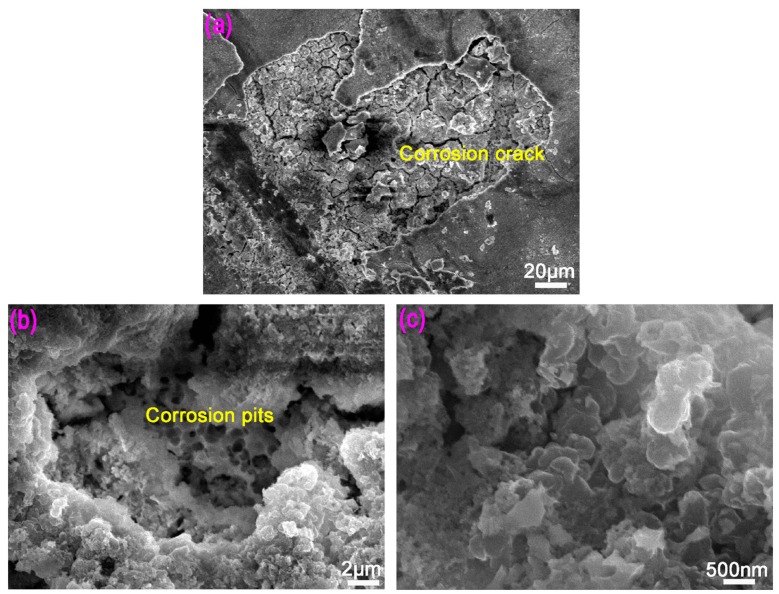
The corrosion view of Mg_60_Cu_23_Gd_11_Ag_6_ glass: (**a**) Overall surface morphology; (**b**) morphology of corrosion pits after removal of corroded surface layer; and, (**c**) number of particles inside corrosion pits.

**Figure 8 materials-12-02989-f008:**
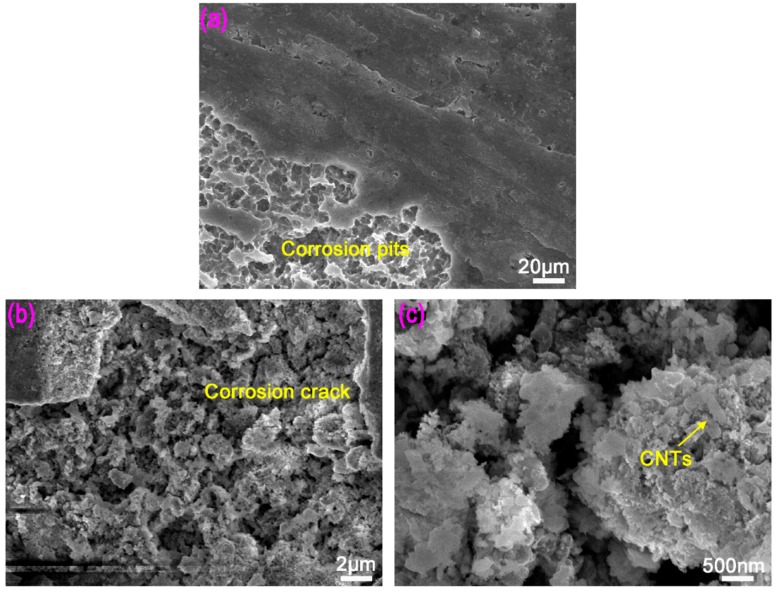
The corrosion picture of carbon nanotubes reinforced Mg_60_Cu_23_Gd_11_Ag_6_: (**a**) Overall surface morphology; (**b**) morphology of corrosion pits; and, (**c**) pattern of snowflake corrosion grains coated with CNTs.

**Figure 9 materials-12-02989-f009:**
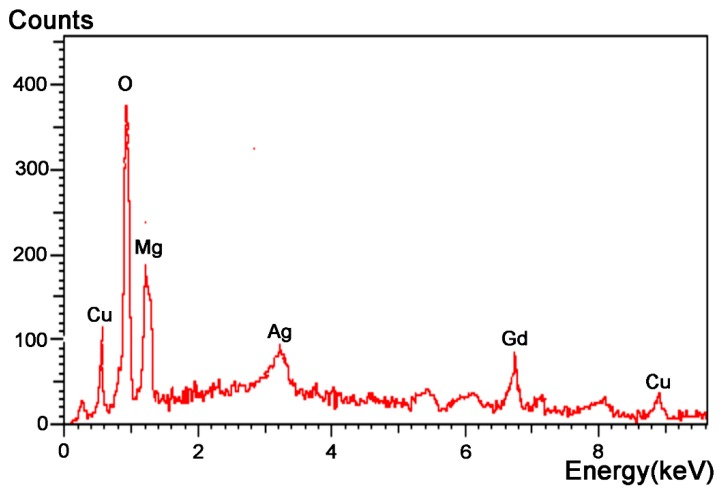
Energy dispersive spectrometry (EDS) analysis of carbon nanotubes reinforced Mg-based bulk metallic glasses products after corrosion in 0.1 mol/L NaOH solution.

**Figure 10 materials-12-02989-f010:**
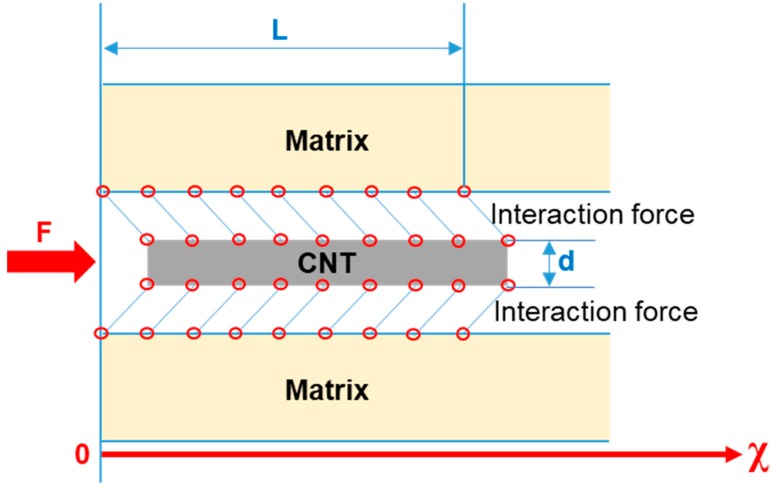
Schematic diagram of shear-lag model for the interaction between the carbon nanotube and the metal matrix.

**Table 1 materials-12-02989-t001:** Mechanical and physical properties data of Mg_60_Cu_23_Gd_11_Ag_6_ and CNTs-Mg_60_Cu_23_Gd_11_Ag_6_ bulk amorphous alloy rods.

Sample	Compressive Strength, σ_c_ (MPa)	Fracture Strain, ε (%)	Density, ρ (g/cm^3^)	*σ*_c_/*ρ* (kN·m·kg^−1^)
Mg-BMG	812	1.91	4.31	188
CNTs-Mg-BMG	1007	2.67	4.10	245

**Table 2 materials-12-02989-t002:** Contents of elements in CNTs-Mg-BMG products after corrosion in NaOH solution of 0.1 mol/L.

Elements	O	Mg	Cu	Gd	Ag
Atomic content	56.10	16.16	11.29	10.86	5.58

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
