# Peer review of "The Significant Impact of Carbon Nanotubes on the Electrochemical Reactivity of Mg-Bearing Metallic Glasses with High Compressive Strength"

_materials, 2019, doi:10.3390/ma12182989_

Round 1

Reviewer 1 Report

There are several studies in the literature of CNT reinforced BMGs including Mg-BMG. Authors must elaborate further on what this specific study sets out to do that previous research work hasn't. Provide citations on page 4 where it is stated "a large number of experiments confirmed that the ...".  Bottom of page 5 - authors state that compressive stress-strain curve of Mg-BMG is pre-dominantly elastic while that of CNT-reinforced alloy is elastic-plastic. However both stress-strain curves appear to show catastrophic brittle failure following elastic deformation. There doesn't appear to be a distinct yield point in the stress-strain curve for the CNT-BMG - how has this yield point been determined? In addition, what is the reason for change in slope in both stress-strain curves at roughly 200MPa?  Provide error bars while reporting measured data Top of page 7 - has delta T due to adiabatic heating been estimated or calculated? On what basis do the authors make the following claim : "The adiabatic heating ΔT was generally higher than the glass transition temperature of Tg, but if it was lower than the solidus temperature Tm, plasticity at the macroscale may not occur and the fracture would be brittle rupture"? Has the theory presented on Page 12 been used to estimate stresses in the CNTs and calculate pull forces reached at their failure points?

Reviewer 2 Report

Paper can be accepted after the following corrections:

- Please avoid acronyms in figure's captions to increase clarity.

- Figure 2 should be presented accordingly to the scientific standards. Please explain the key elements of the devices (numbering them 1, 2, 3 etc.) as well as indicate the unique solutions in each device (if any). Please also clearly specify the type of each device, if commercially available.

- Strain scale should be clearly presented on x-axis. Please split into sub-figures if necessary.

- Figure 10 is very important for the results and should be clearly presented. Please re-draw accordingly to scientific standards. Please also clearly presents the interaction between the CNT and the metal matrix. Right side sub-figure is trivial and can be omitted.

- Equation in the section "Conclusion" should be re-moved or moved to text.

Round 2

Reviewer 1 Report

Thank you for providing clarifications and updates to the points raised previously.

Reviewer 2 Report

Paper was corrected and can be accepted in the present form.